# Phage libraries screening on P53: Yield improvement by zinc and a new parasites-integrating analysis

Sihem Ben Abid ☉*, Emna Ketata ☉©, Ines Yacoubi ☉¤©, Lamia Djemal, Salma Abdelmoula-Souissi, Aida Koubaa, Raja Mokdad-Gargouri ☉, Ali Gargouri ☉

Laboratoire de Biotechnologie Moléculaire des Eucaryotes, Centre de Biotechnologie de Sfax, Université de Sfax, Sfax, Tunisia

© These authors contributed equally to this work.
¤ Current address: Laboratoire de Biotechnologie et Amélioration des Plantes, Centre de Biotechnologie de Sfax, Sfax, Tunisia
* sihem.abid@gmail.com

**Data Availability Statement:** All data generated or analysed during this study are included in this published article [and its supporting information files]. The datasets generated and/or analyzed

## Abstract

P53 is a transcription factor that controls a variety of genes, primarily involved in cell cycle and other processes related to cell survival and death. We have isolated peptides targeting P53 (protein and domains) using the "phage display" technique. Interestingly, adding ZnCl2 at 5–10 mM in panning solutions helped to recover more plaque-forming units at least at round one of the screening. Subtractive docking analyses were designed by using a pool of common redundant peptides known as *parasites*. This rationale helped us differentiate between possibly specific and non-specific bindings. We found notable differences in docking characteristics between different sets of peptides either related to different targets or related to zinc-conditions. The set of zinc-related peptides shows advantageous docking profiles: sharper binding for some positions and distinct exclusive bound residues, including the relevant R248 and R273. Zinc would have modulating/helping role in the targeting of protein P53 by phage displayed peptides in addition to an enhancement action on bacterial infection.

## Introduction

The P53 tumor suppressor protein is involved in the cell cycle control and the DNA repair mainly as a transcriptional factor targeting genes involved in these processes [1]. In some cancers, mutations affecting p53 alter several of its activities [2, 3]. In addition, P53 is implicated in neurodegenerative diseases; it interacts with tau, specifically the oligomeric tau, and forms oligomers and fibrils in human Alzheimer's disease (AD) brain [4]. P53 operates at the nuclear and the cytoplasmic levels in several processes and in particular in various cell death pathways such as apoptosis [5] and autophagy [6]. All this makes P53 a major target for research and pharmacotherapies. Peptides are one of those therapeutic molecules that have been tested on P53 with varying degrees of success. The Phage display represents a reliable source of such

during the current study are available in the [figshare] repository, https://figshare.com/s/e94677911829f955999e. 10.6084/m9.figshare.24045519.

**Funding:** SBA 2 Grants Ministry of Higher education and scientific research- Tunisia http://www.mes.tn/index.php Center of Biotechnology of Sfax- Tunisia http://www.cbs.rnrt.tn/ The funders had no role in study design, data collection and analysis, decision to publish, or preparation of the manuscript."

**Competing interests:** The authors have declared that no competing interests exist.

peptides. After a first basic essay made by Daniels and Lane [7], Tal et al developed a higher throughput phage screening on the p53 protein, by sequencing a great number of peptides [8]. Some of them showed potency on tumor regression in mice. The same research team developed some lead peptides among numerous isolated sequences towards clinical mutant-p53 reactivating application [9]. We also made a contribution consisting in the selection by phage display and study of a heptapeptide called NG7 [10].

Targeting approaches can vary from a direct effect on protein structure-function relation to an interfering binding with partner(s). For wt-p53, researchers have commonly sought to enhance the activity of the protein by dissociating it from negative regulators, thus leading to the accumulation of the functional protein [11, 12]. These negative regulators are mainly the E3 ubiquitin ligases MDM2 and MDMX which are causing the degradation of P53 by the proteasome [13]. This applies for example to the PNC-27 anticancer peptide family [14]. Phage display has been a valuable tool in the targeting of this particular axis P53/MDM2-MDMX.

Inhibiting or diminishing the protein aggregation is another approach which can concern both wt and mutant p53 [15, 16]. Researchers used peptides based on aggregation prone sequences in the targeting of the tumor suppressor p53 protein [17]. The ReACp53 designed peptide caps an exposed sequence prone to aggregation on the protein, inhibits mutant p53 aggregation and restores it [18]. For the mutant forms, usually approaches aim to stabilize the protein. This is the case for the CDB3 peptide derived from the p53 partner 53BP2 which stabilizes the core domain of the mutant protein [19]. Researches also have sought to prevent the oligomerization/tetramerization which can lead to hetero-oligomers composed of wt and mutant monomers.

Another structure-function feature widely used in P53-targeting strategies is the zinc binding property. We recall here that a zinc molecule is maintained by four bonds with four residues of p53 protein which are important for its binding with DNA [20]. Strategies vary from supplementing non functional P53 proteins with Zn ions to using metallochaperone proteins in regulating zinc concentrations [21, 22]. However, to our knowledge no existing results about peptides targeting P53 with relation or depending on the zinc presence or state.

Furthermore, and though the technique of phage display screening has brought real advancement to the targeting strategies and provided interesting ligands, it has nevertheless yielded in some cases non specific interactions. One special category of them often retrieved by separate works was characterized and known as *parasite* phages [23, 24].

We sought in this study to select small peptides having an affinity with human P53 using phage display libraries screening. The ultimate goal is to find new peptide-ligands which could modulate some of the functions of the protein in further future tests. To analyze the results of molecular docking of isolated peptides with protein P53, a rationale of subtractive binding is conceived and followed taking advantage of an obtained pool of redundant *parasite* phage clones. Above all, interesting effects of zinc supplementation during screening and titration are highlighted.

## Results

### Zinc chloride addition increases the phage titers at the first round

Regarding phage display against the P53 protein, Zinc chloride (ZnCl2 5-10mM) was added during different panning steps, knowing that zinc ions are necessary to the p53 protein to maintain a good conformation and therefore a good functionality. Table 1 shows the titers obtained (pfu/μl) in round 1, depending on whether or not ZnCl2 salt was added. As a consequence of this presence of ZnCl2, we noticed for this round 1 an increase in the number of phages (plaques) obtained: ten and seven times more for the 7 mer and the 7 mer C-C

**Table 1. Phage titers at round 1 of screening on GST-P53 protein.**

| Condition | -Zn | | +Zn | | A1/E1 ratio | | +Zn/-Zn ratio | | |
|---|---|---|---|---|---|---|---|---|---|
| Library | E1 | A1 | E1 | A1 | -Zn | +Zn | E1 | A1 | A1/E1 |
| 7 | 103 | 2.1E+05 | 1.0E+03 | 3.0E+09 | 2.0E+03 | 3.0E+06 | 10 | 1.4E+04 | 1.5E+03 |
| 7C-C | 1 | 1.0E+03 | 7 | 1.0E+10 | 1.0E+03 | 1.4E+09 | 7 | 1.0E+07 | 1.4E+06 |
| 12 | 1 | 1.3E+05 | 1.0E+03 | 1.0E+11 | 1.3E+05 | 1.0E+08 | 1.0E+03 | 7.5E+05 | 7.5E+02 |

Phage titers (pfu/μl) of the first-round *Eluates* (E) and *Amplifiates* (A) of the three phage libraries. Ratio A1/E1 which reflects the amplification rate is deduced. Libraries are 7 mer (7), 7 constrained mer (7C-C) and 12 mer (12) sized. (Cond: conditions, Lib: libraries).

respectively, but especially for the 12-mer peptides library ($10^3$ times). This improvement concerned the eluates in the first round.

This positive effect of zinc treatment was confirmed with the amplifiates (A1). The A1 titers are much higher for the "*with zinc*" condition than for the "*without zinc*" condition and this is true for all three libraries. Finally and for the three libraries, the amplification rate (given by A1/E1 ratio) is higher when adding zinc compared to the standard (Table 1). The two other rounds of panning were not titrated; first because it was not aimed to and this zinc effect on phage numbers was observed by chance. Titration is habitually realized with (recommended) serial dilutions which may substitute to repetition.

## The constrained peptide library has low titers at elution and Zinc chloride barely changes it

With regard to 7 C-C library, numbers of phages recovered at this eluate level are too low with or without addition of zinc; that we cannot conclude on a notable effect of ZnCl2. Interestingly, after amplification, there was an important increase in phage titer and this increase is more pronounced with zinc addition.(Table 1).

We think that there were more phages in the eluate of the C-C library than assessed by titration. On the other hand, the low titer of the eluate (E1) of dodecamers' library, under standard condition, can have different explanations resumed in: technical defects. This will have no repercussion later in the amplifiate titration due to the "exponential" nature of phage multiplication.

## Screening results

**New P53-recognizing peptides.** Twenty distinct motifs resulting from the screening on the whole GST-P53 protein were obtained: eight patterns (7-mer) and twelve patterns (12-mer) of the correspondent library types (Table 2A and 2B). We mention that sequence 7.3 (NPNSAQG) also appears in our previous publication [10]. Phage display against p53-derived domains gives extra motifs: seven 7-mer sequences for the PD74 peptide (Table 2C), and only two 12-mer's for the SR50 (Table 2D).

**Screening giving redundant motifs: "Parasite phages".** Other motifs recognizing either PD74 or SR50 or both were separately found, using whether the C-C library or a mixture of the three libraries together: 18 sequences that we called: R0 till R17 (Table 2E). The remarkable fact was the great similarities between these motifs. However, two consensuses can be sorted out and are shared by the two targets/screenings: NGVEIPP (or GVGIPL) (7-mers) and PFNEPHL (C-C 7-mers).

Some redundant "R" motifs have been already sorted out by our laboratory colleagues in distinct screening works. Researchers, as advanced in introduction, have already reported this

**Table 2. Phage titles and *lists of P53 recognizing peptides*.**

| a: Set P53 7 mer | | c: Set PD74 | e: R set | |
|---|---|---|---|---|
| | | | Motif 1 (7) | Target |
| **Non Zinc** | **With Zinc** | PD1: GANMKYA | R0: VGVRIPL | PD74 |
| 7.1: HTWLRSA | 7Z1: WSWPRFL | PD2: GLTATNM | R9: PGVGIPL | SR50 |
| 7.2: LHNSLPA | 7Z2: MQAPSPM | PD3: GFTATNM | R10: IRVGIPL | SR50 |
| 7.3: NPNSAQG* | 7Z3: AAAFTQS | PD4: NDAEMPT | R8: VGVGIPP | SR50 |
| 7.4: ATHQTLR | 7Z4: GTEPPAM | PD5: ETTHARA | R1: NGVEIPP | PD74 |
| | | PD6: GLDCYKQ | | |
| | | PD7: STQARTP | **Motif 2 (7 C-C)** | **Target** |
| | | | R2: PFNEPHL | PD74/ SR50 |
| | | | R3: PFNEPHP | PD74/ SR50 |
| **b: Set P53 12 mer** | | **d: Set SR50** | R4: PINEPHP | SR50 |
| **Non Zinc** | **With Zinc** | SR12.1: HLAQTASPPAAP | R5: PKNEPHP | PD74 |
| 12.1: NHMNQISFPSRP | 12Z1: SHVPLARWSVIT | SR12.2: APLYSPSHLATS | R6: SFNEPHP | PD74 |
| 12.2: ARSPCQVQSRTS | 12Z2: HDHLIPFYWADL | | R7: SINEPHP | PD74 |
| 12.3: NNLAFYHTFISP | 12Z3: STLVFPAHTRDY | | R11: LFNERHP | SR50 |
| 12.4: APSPFQVQSRTS | 12Z4: TYLLPHSYPWYG | | R12: PINEPHL | SR50 |
| 12.5: NYPSSSVPHAPQ | 12Z5: TATLDMPLSLPS | | R13: AFNEPHP | SR50 |
| 12.6: YSTHDNARPWLL | 12Z6: WMDSYMSQHDWP | | R14: AINEPHP | SR50 |
| | | | R15: AINEPHL | SR50 |
| | | | R16: AFHEPHP | SR50 |
| | | | R17: AIHEPHP | SR50 |

(a-b) Lists of peptides recognizing the P53 protein, of 7-mer and 12-mer sizes respectively. (*) Sequence 7.3: NPNSAQG is reproduced from [9]. (c-d) Lists of peptides recognizing p53 domains (PD74 and SR50 respectively) and (e) List of redundant motifs (R).

type of result "obtaining very frequent repetitive motifs, either in the same screening or in distinct target screenings" [25]. These sequences are known as *Parasites*.

## Docking analysis

A first parameter of docking considered a result is the energy. Values are resumed for all peptides in (S1 Table). And we can already remark the superiority of redundant "R" sequences on the others.

The other parameter is the docking/binding profile which is a schematization that can yield informations on the interactions between the identified peptides and the P53 protein at the sequence level. This can be made either at an individual level (peptides) or for sets of peptides (to find out some common features of interaction, patterns).

**Binding profiles in docking to 2LY4.B.** Binding of peptides to 2LY4.B structure has an overall profile of two regions separated by a small area of non-binding in the center A39-M40. The binding frequency ranges from low to high from the ends toward the vertices of these two regions. This is almost the same profile for R set of redundant peptides. Fig 1 shows a comparison once between the 7 and the 12-mer sets (size) and another between the *non zinc* and the *with zinc* states (zinc status) and this is done for the initial form (including the redundant residues) and by subtracting them.

*The ZnCl2 effect on docking profiles.* Considering the 7-mer sets, the « *non zinc* » profile is closer to that of the *R* set than to that « *with zinc* ». For example, two high contact points S33 and L43 shared between the *non zinc* and *R* sets are weakly present, in the *7 with zinc*. (Fig 1A)

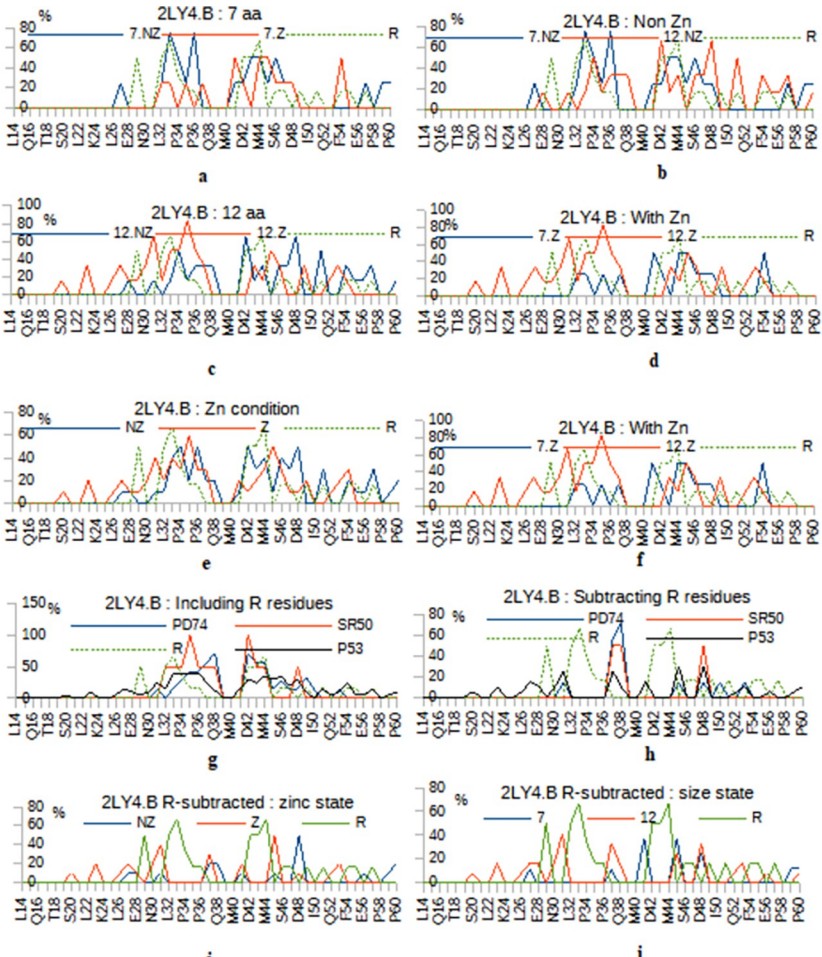

**Fig 1. Docking profiles of peptide sets with 2LY4.B.** Two ways: including the redundant (R) residues **(a-g)** and with R subtraction **(h and j)**. Comparing the zinc states **(a, c and e)**, comparing the two sizes **(b, d and f)**. And comparing separately the zinc states and the size states on R subtracted condition **(i and j)**.

ZnCl2 treatment in the 7-mer phage screening made both L43 and P27 buried. It also made the F54 residue newly exposed or available while masking the D57_P60 end reached when docking standard peptides.

For the 12-mer group, the main effect of the ZnCl2 treatment was to *promote contact with aliphatic/hydrophobic residues* like L45 made accessible as with L35 and V31 or even W23, upstream on the sequence. At the C-terminal end, the zinc treatment, again, *enhances the neutral/hydrophobic residues* (Q52 and W53) and *disadvantages the acidic ones* (E51 and D57). (Fig 1C).

In conclusion, *less charges and more aromatics with the ZnCl2 condition but that was before the R correction*. Moreover, we can observe how the zinc condition refines the docking interactions, especially for the first part of the structure, and makes the contacts sharper and more punctual, also adding more binding-sites.

Redundant bonds frequencies vary from low ~17% (region 46–56) to high (33, 50 and 67%) in region (29–45).

If we correct by subtracting the common residues from the set *R*, there remain fewer contacts; they flank regions of strong redundant binding or alternate less frequent but ubiquitous

bindings. In terms of distribution, differences can be observed between all sets. While the 7-*mer non zinc* positions are spread out (from 27 to 61), the *7-mer with zinc* has only four close contacts at S37, D41, L45 and D45, thus a localized binding. We also observe that "*12 non zinc*" which covers almost the same region than "*7 non zinc*" (middle and C terminal), has a shift compared to "*12 with zinc*" which prefers to bind to the N terminal of the structure. We cite S20, W23, P27 and V31 and far downstream, W53 (Fig 1H–1J).

Finally, by clearing all sets from R residues but also SR50 few positions (non specific), we obtain for the total set of peptides targeting the fixed P53 protein, the following positions in decreasing order of frequency: L45 (30%), V31 (25%), P27 and D41 (15%), W23, E28, N30, W53 and P60 (10%), S20, L26, Q52, E56 and G59 (5%). With residues of the *non zinc* pool being: E56, G59 and P60, those from the *with zinc* pool being: S20, W23, L26, N30, Q52 and W53 and those common to both being: E28, P27, V31, D41 and L45. (Fig 1H) The *PD74* set now binds 2LY4.B on four contacts : V31, L45, I50 and W53 all present at 14%. Details of docking structures are presented in (S1–S5 Figs).

**Binding profiles in docking to 3Q01.** *Some relevant residues are exclusive to the "plus zinc" set.* Fig 2 shows the docking profiles of all peptide sets whether including the R redundant residues or after subtracting this redundancy from interactions. On the DNA Binding Domain (DBD), the interaction of all sets recognizing P53 protein can be broadly summarized in three regions: 96–165, 200–233 and 248–291. Considering the docking frequencies, an ascending order is observed: "*12 non zinc*" (16,67%) then both 7-mer sets at 25% but the "*7 with zinc*" has higher binding at some points. And finally, the "*12 with zinc*" set binds at 20%, 33% and 50%. (Fig 2).

After *R correction*, only few contacts are lost for "*12 with zinc*" and these are the least frequent binders. The "*12 non zinc*" set loses many binding points. For the 7-mer's, it is the "*with zinc*" set which loses the relevant binding points which are the most frequent.

When comparing the docking profiles of the 7-mer sets with the two 12-mer sets, differences are obvious (Fig 2A–2D). Note that the former only bind with interval (97–267). The 12-mer sets bind everywhere but seem to favor, in particular the "*12 with zinc*", the region (248–291) in the extreme DBD.

The 7-mer sets of « *Non Zinc* » and « *With Zinc* » nearly coincide in docking positions except for a few offsets or few distinct contacts. The difference between the two zinc- conditions is clearer for the 12-mer size.

Fewer binding positions are observed for "*12 With Zinc*" especially in the DBD. However, these docking points are stronger than those of "*12 Non Zinc*": with higher frequencies and reinforced by doublet's interactions. (Fig 2A–2D)

For the « *Non Zinc* » pool, 7-mer and 12-mer almost coincide in bound sub-regions. For the «*With Zinc* » pool, the shift of the 12-mer set to the right (the C-terminal part) compared to the 7-mer is evident.

For SR50 and after subtracting *R*, the docking concerns only a few contacts that all except E271 are probably non-specific since outside the SR50 location. We noticed that the SR50 set shares almost all of its few positions with *12 with zinc*. There is possibly a similarity between the conformations of the synthetic SR50 and the same peptide in the « *With Zinc* » P53 protein; zinc would have exposed some regions there (Fig 2G and 2H).

The PD74 group (negative control here) shows low binding frequencies for most involved residues in the DBD. The frequency (~14%) is a little higher than those of R set. When R- corrected, only a few contact residues remain which still can be considered as non-specific (Fig 2G and 2H).

For the C-terminal region, after clearing, most of the binding contacts are probably non specific/exclusive: found with *R* set or *PD74* or *SR50*; a single (specific) residue (A353) can be retained for the 12-mer set.

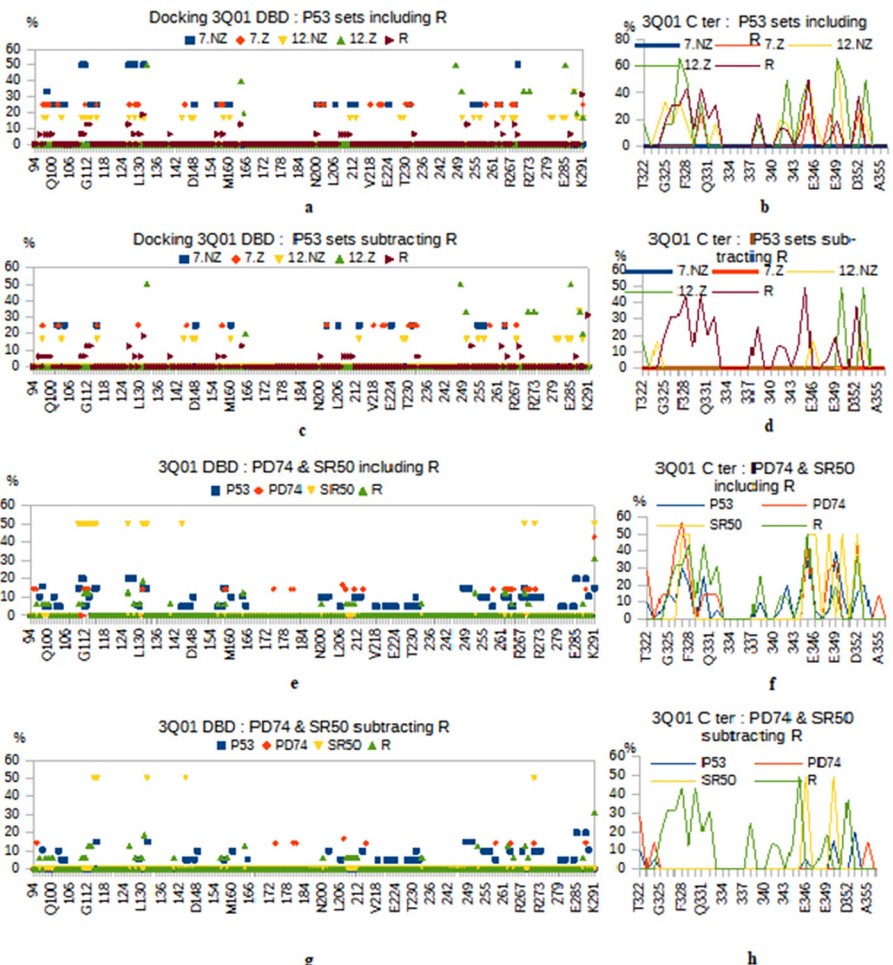

**Fig 2. Docking profiles with P53 structure 3Q01.** Profiles related to the DBD (left) and the C-ter (right) separately. Sets: the P53 targeting sets, once including common residues with redundant set R (**a-b**) and once subtracting them (**c-d**), and for the P53-domains targeting sets, including R residues (**e-f**) and subtracting them (**g-h**).

Some residues are rather exclusive to the zinc condition: A129 (7-mer), K132, Q165, R248, P250, E271 and R273 (12-mer). Thus, the relevant residues like the R248 or R273 hotspots are finally reached. We even notice that this binding occurs through the doublets 129/132, 248/250 and 271/273 which means that it is rather strong. R248 is bound by 12Z2 (by W9) which also connects P250 by Y8 and R273 by A10. 12Z3 binds R248 (by L8) and R273 (by S1) which binding is consolidated by E271 bound by D11. This 12Z3 has an interesting binding involving both the DBD and the C-terminal region; the latter reached by the trio F5-P6-H8. 12Z5 binds R248 via its L3; it also connects to the C terminal region via its L10-P11 pair. Details of docking structures are presented in (S6–S19 Figs).

**Comprehensive analysis of docking and study of individual peptides.** For a comprehensive view of the zinc effect in modulating the binding characteristics of the peptides to P53, we resume as follows: with the N-terminal domain docking, there is almost a superposition of binding points between the « Non zinc » and the « With zinc » sets. Residues L35 and L45 are the preference of the « With zinc » set while D48 is preferred by the « Non zinc ». For the C-terminal part from the docking on 3Q01 structure, the most of the binding is ubiquitous non specific. Except these close positions L350 and A353 which may be specific and the L350 preferred by the «With zinc».

**Table 3. Retained residues versus subtracted following R correction.** (Str : structure).

| Str | Set | Remaining residues | (R) Subtracted residues |
|---|---|---|---|
| 2 LY4.B | 7.NZ | H. L **W**.. (A/R) | (L/A) (H/P) N S A (P/Q/L) . |
| | 12.NZ | . Y (A/P). Q. . . (F/S/R) (I/R). Q | (A/N/Y) (H/R) (S/T) (S/H) (F/S) (Y/Q) (H/S).. **W**. (P/S/L) |
| | 7.Z | **W** (S/Q) **W** F R. S | (M/G) (A/T) (A/E) P. Q M |
| | 12.Z | (T/**W**) (H/Y/M). P Y P . **W** (P/**W**) . **W** (T/Y/P) | (S/H) (D/T) (H/L/D) L (L/P) (M/H) (S/P) Q S (A/**W**/L) R L |
| | PD74 | N (L/F/D) Q .. P (M/T/P) | (G/E) T (N/T/E) (M/A) R (N/Y/K) A |
| | SR50 | …..… H.. T S | H (L/P) (A/L) Y T P S P L.. P |
| 3 Q01 | 7.NZ | (H/L). (**W**) (S/L) (L/R/T). R | (N/A) T (N/H) Q A (Q/L) (A/G) |
| | 7.Z | **W** (A/T/S) (**W**/E) F (P/T/R). (L/S) | M Q P. S P M |
| | 12.NZ | Y (H/S) T (H/N) D. (S/H) (F/T/Q) P. (R/P) (P/L) | N (N/Y) (L/S) (A/S) (F/S/C/Q) (Q/Y) V. H R T (S/Q) |
| | 12.Z | S A.. F P F (**W**/H/L) (**W**/H) (L/R/A) (D/P/**W**) (Y/S/T) | (T/H/**W**) (H/D/Y/M) (L/V/M/H) (L/S) (P/I) (A/M/H) (A/S) Y.. Y G |
| | PD74 | . (A/N) N (M/E) K Y (A/P) | (G/E) (L/F) (T/A/D/Q) (A/C) (Y/R) (N/P) (M/Q) |
| | SR50 | .. L Y.. S. P. A P | H L A Q S P S H.. T S |
| R motif 1 | | | (V/N/P/S) (G/I/F) (N/V) (E/R) (P) (P/H) (P) |
| R motif 2 | | | (P/A/V/N) (F/IG/K/R) (N/H) (E/G/R) (P/R/I) (H/P) (L/P) |

The most of the effect thus concerns the DBD; S20 Fig was added to show the docking of the two zinc state sets on this region. Apart of some common positions between the two zinc sets, there are subregions specific to one or the other set. We notice regions 129–134, 219–223, 248–250 and 272–275 reached only by the "With zinc" set. The binding made by the "With zinc" set reaches frequencies higher than the "Non zinc" set (till 30% of peptides set) indicating probably good docking points for certain positions. This concerns K132, R248, E285 (at 30%) then E271 and R273 at 20%. Meanwhile, we notice that some "With zinc" bindings such as 129–134, 156–159 and 164–167, occur close to regions of redundant ubiquitous binding. Some high frequent binding points from the "Non zinc" set are absent from the "With zinc" docking, citing D148, M160, R213 and I254/T256. To conclude, we need more sequences to analyze from both conditions in order to come with stronger conclusions.

Details of retained positions and subtracted ones for the distinct sets of peptides are summarized in Table 3.

We notice the presence of more Trp residues in the retained sequences than in the subtracted positions. These Trp residues overabundance can result from a bias as described in [26]. It has been advanced that this Trp bias in phage display output indicates that the corresponding positions are important to the interaction with peptide ligands [26].

At the individual peptides level, a representative ELISA test is shown by S21 Fig. Docking profiles of individual peptides after R correction are presented in S22 Fig and analyzed in S1 File.

## Discussion

We performed p53 targeting by phage libraries. As a zinc-binding protein, P53 binds a zinc molecule and this leads to a conformation change which enables it to bind DNA. That was the main motivation to test the addition of ZnCl2 salt in screening solutions. Interestingly, the addition of ZnCl2 proved effective in improving the number of pfu obtained for eluates at round 1. Reasons for this phage titers improvement can be variable. There are globally two hypothesis : either the Zn enhanced the obtention of more phages through the panning steps

or it helped enhancing the infection of bacteria during the titration. It has been found that metal cations: iron, magnesium, and calcium enhance the phage killing of bacteria [27]. Those researchers gave a rich discussion about possible explanations for this effect of metal cations (Fe, Mg and Ca) on phage infection and killing of bacteria. This can probably apply on Zn cations and we think that Zn would have improved phage infection and/or stability at the titration steps. For the amplifiates, the Zn enhancement is relatively less important than for eluates since ZnCl2 was already diluted.

This implies that the efficiency of the screening process, in terms of representativeness, can be improved by such (ZnCl2) treatments. And, there are phage clones that may be lost at the standard method, without (ZnCl2) salt addition.

Normally, amplifiates should have higher titers than eluates at each round of phage display screening (~$10^7$ times according to technical manual). Here, in the absence of zinc, both titers are lower than normal. The presence of zinc partially redressed the titers. This starts with the eluates' titers. The amplifiates' titers just follow, the amplification is exponential like. At elution, the effect of zinc is more remarked for the 12-mer library then for the two 7-mer libraries. Thus, a possible limitation against long protruding peptides exposed on phage pIII protein can exist, in standard conditions of screening. This is not only eliminated by zinc, but also reversed (the higher amplifiate titer). Similar limitation reversion, due to zinc presence, could concern the constrained 7-mer peptides, but more at the amplification step level. Many other sequence biases can occur during phage amplification as described by a large literature, this includes the parasite phages appearance.

The amplification rates reflect the multiplication of eluted phage pools by bacterial infection; here they are almost comparable to what we can find elsewhere in other screening experiments. We think that, in absolute, the amplification rate itself and its variation tendency through screening rounds, can not be really taken as an evaluation of the enrichment of the obtained clones or even of the success of the screening, since the redundancy (parasites) can still be recovered. And parasite clones tend to be more amplified than other clones according to literature. Here, the amplification rates permitted to confirm the improvement caused by zinc salt, already observed at elution. Only sequencing, sequence analysis including comparison to known parasites, and functional tests can orient us toward a potential utility of any obtained phages.

The presence of metal ions (ZnCl2) was not sufficient enough to enhance infection at titration for C-C library eluates. With or without ZnCl2 in solutions, eluates titers were low while amplifiates were much higher for both conditions. The low eluates titers probably don't reflect low phage charges. The constrained C-C phages were obtained in a form/state which can not infect bacteria properly following elution. But this changed after amplification. And we retrieved expected important titers, yet again with a superiority for the "ZnCl2" pool.

At another level, Zn presence would have led to conformational changes that occur on the P53 protein, specifically by exposing more binding sites or residues including R248 and R273. This may provide a piece of explanation to the fact that supplementation of some p53 mutants (R175H and R273H) with zinc recovered their functionality [21] as well as to the zinc metallo-chaperones capacity to restore the p53 and recover its functions [22]. The important region for MDM2 binding on p53 is known to be (15–29), with the triad (F19, W23 and L26) forming the amphiphilic α-helix in the complex [28]. Peptides 12Z6 and 12Z3 both reaching residue W23, and 12Z1 by binding tripeptide 26–28, are good potential interfering ligands. Numerous data support a crucial role of zinc in "conformational" regulation of P53 [29, 30].

At our knowledge, this is the first time, zinc addition was tested in a phage display screening against P53. Results are interesting. Therefore, adding zinc in peptide screens and tests can become a systematic choice, as well for P53 and for other targets.

In the N-ter structure, we can notice special features such as a symmetry of binding around residue L45, common to *non zinc* and *with zinc* categories however reversed between them.

This symmetry engages between others the ubiquitous positions since it disappears when R corrected. Researchers have stipulated a mechanism of interaction between transactivation domain (TAD) of the P53 protein and its partner the replication protein A (RPA), which consists in a conformational transformation of this TAD upon binding. The originally disordered TAD in solution, folds via its residues 37–57 into two amphipathic helices, H1 and H2 [31]. Moreover this TAD conformation seems to be modulated by some ligand binding and cancer-associated mutations N29K/N30D [32]. These effects are mainly achieved through nonspecific interactions [32]. Here, it can be said that this domain can have two distinct behaviors in the absence and in the presence of zinc ions. We believe that peptides screening and binding studies can bring some understanding to this structure-function knowledge of the TAD region.

Either for P53 or other targets, the possibility of identifying redundant motifs should be taken into consideration. The good study by Sloth et al [25] have identified a list of the most common "parasite sequences" that researchers have isolated sofar. They ensure that some of these sequences show functional effects on targets and thus they are not « totally » non specific. We might isolate, in a screening, either some of known parasite sequences or completely different or mixed sequences partially composed of parasites. The challenge is then to test the degree of efficency of isolated peptides and to rank them based on « specific action ».

This heterogeneity (parasites/non parasites) probably affect the reliability of the screening results, for P53 as for other targets. Censorship and bias of the constructed libraries are thought to be the origin of these parasites. As literature suggested possible solutions, the high throughput sequencing and avoiding biased libraries or equilibrating biased ones.

The obtention of parasitic redundant/ubiquitous motifs and their docking profiling made us choose to substract their interactions from the rest as a "noise" correction. Retained interactions remain interesting even if they are few in number. Further confirmation with techniques like alanine substitution and biophysical binding studies (RMN. . .) will help us assess the accuracy of this rationale (of parasite subtraction).

In summary, we describe a new approach to refine the screening of peptides potentially applicable to other targets. In conclusion, we have provided new insights into the phage library screening method.

## Materials and methods

### Strains, vectors, primers

- ER2738 (New England Biolabs) *E. coli*: *F´proA+ B+ lacIq Δ(lacZ)M15 zzf::Tn10(TetR) / fhuA2 glnV Δ(lac-proAB) thi-1 Δ(hsdS-mcrB)5*. Used as host in phage display experiments, the supE suppressor allows the read-through of TAG stop codons.

- BL21 (DE3) (Stratagene): *E. coli* B, F⁻, *dcm*, *ompT*, *hsdS* (r$_B$⁻m$_B$⁻), *gal* (DE3), protease deficient and contains the T7 RNA polymerase gene, under the control of the *lacUV5 promoter*, integrated in the bacterial chromosome. This strain was used for the expression of the GST-p53 fusion protein as for expressing only GST as control.

- vector pGEX-4T-3 (Addgene_79149) to express only GST protein for alternative screening.

- recombinant pGEX-4T-3-P53: The human-p53 cDNA was previously cloned in the *EcoRI* site of pGEX-4T3 under the lac-trp hybrid promoter that is inducible by IPTG, and fused upstream to the GST (Glutathione S-Transferase), see [10].

- Primer used for the sequencing of phages clones: -96 gIII sequencing primer: 5 ´-CCCTCATAGTTAGCGTAACG, set at 1 pmole/μl.

## Peptide synthesis

Two peptides spanning two parts of the p53 coding sequence were synthesized by ProteoGenix (Schiltigheim, France): PD74: `PPLSQETFSDLWKLLPENNVLSPLPSQAMDDLMLSPDDIEQW FTEDPGPD` is a 50 mer covering the acidic N-terminal region (residues 12–61 of p53), with a MW of 5670.3 Da. SR50: `SCMGGMNRRPILTIITLEDSSGNLLGRNSFEVRVCACPGRDRRTE EENLR` is a 50 mer covering a part of the DNA-binding central domain (residues 241–291), with a MW of 5624.4 Da.

**Rationale for target selection.** The PD74 peptide correspondent to the N-terminal part of p53 (12–61) contains many relevant motifs such as the important 9aaTAD. These 9 amino acids transactivation domains are thought to have autonomous transactivation activity in some eukaryotes and may also mediate conserved interactions with general transcriptional cofactors [33]. And SR50 originated from the central DNA-binding domain (residues 241–291) containing many important residues and mutation hotspots.

## Target preparation for phage libraries screening

**P53 protein purification.** Protocol was that used in [10] with small modifications. Cultures of BL21 *Escherichia coli* bacteria strains expressing separately the GST and the GST-P53 fusion were prepared as following: culture was conducted at 37°C until an $OD_{600nm}$ of 0.9, and then induced by 0.5 mM of IPTG for 2 hours at 30°C and at 250 rpm. After lysis with osmotic method, total protein extracts (soluble fraction) are mixed with Glutathione sepharose beads. Beads are then extensively washed to eliminate unbound proteins and particles. Three pools of GST-P53 fixed beads are used for each type of phage libraries; one per round of screening so that at the end three rounds were done. A pool of GST-fixed beads is prepared and used for subtractive panning which aims to eliminate GST-specific phages in the eluate as well as phages recognizing any other contaminating proteins or particles.

**Preparation of P53 derived peptides targets.** Peptide PD74 was prepared as a solution of 2 mg/ml in water. This peptide showed some difficulties to suspend, despite the addition of $NH_4OH$ and sonication. Dilution of peptide was made in buffer $NaHCO_3$ 0.1M pH 8.6 to have the final work concentrations of 10–50μg/ml for phage display. Peptide SR50 was similarly prepared in a final concentration of 50μg/ml. For coating, 150 μl of diluted peptide solution were used per well.

## Phage display experiment and post-screening studies

**Phage display outlines.** The phage libraries kits used were purchased from NE-Biolabs (NEB), specifically seven-mer linear libraries, constrained (C-C) seven-mer libraries and the twelve-mer displayed peptide libraries. They are constructed in M13KE phage vector with supplied titers of about 2 $10^{13}$pfu/ml. All methods used in this work are based on instruction protocols provided by NE-Biolabs. We cite the following procedures: screening by panning procedure, amplification of phage eluate, collection of phages by using the Polyethylene glycol (PEG)/NaCl procedure, extraction of single stranded phage DNA and physical characterization of positive clones.

Phage titration is performed by one of two methods: bacterial infection on agar plate (as in NE-Biolabs manual) or estimation based on the optical density of the phage preparation (as in phage display book [34]). In the second method, the spectroscopic scanning of filamentous phage solution showed an absorption peak at $OD_{269nm}$. Phage particles per ml = (Adjusted $A_{269}$) x ($6x10^{16}$)/(Number of nucleotides in the phage genome = 7222b for M13KE). The measured $OD_{269nm}$ is adjusted by subtracting $OD_{320nm}$ (the baseline).

**Phage display screening.** Using the NEB peptide phage display libraries manual's protocol; we undertook the screening of three different types of libraries. After the binding step

which lasts between 10 min and 1 h then washing, elution was carried out with Glycine buffer pH2. Eluates are neutralized. Titration is performed at almost every step. Eluate (E) and amplifiate (A) are titrated to appreciate the screening efficiency and the enrichment of specific phages during and between the different rounds.

*Protein P53*. Screening of Phage libraries was performed, in micro-centrifuge tubes, starting with a pre-clearing step on GST beads. Thereafter, each pre-cleared phage library is panned on GST-p53 beads; follow the wash and elution steps. An additional condition was introduced as an optimization of the standard protocol. It consists on adding zinc chloride (ZnCl2 5-10mM) at almost all stages of phage libraries panning: binding (5 mM) and washing (10 mM). This is done for all rounds and all three phage libraries.

*Peptides*. The panning was carried out on immobilized targets: the peptides are fixed separately to the surface of the 96-well plate. Concentration of Tween 20 was increased from 0.1% to 0.5% in the third round of panning, with the aim of increasing the affinity of eluted phages by eliminating non-specific phages which would be weakly bound.

Several phages are collected generally from the third round of each screening (E3). They undergo a first step to verify whether they recognize the p53 target, by ELISA test.

## ELISA tests

ELISA tests are carried out between (peptide, protein) targets and isolated phages as follows: the wells are coated with the concerned P53 peptide or protein GST-P53 priorly solubilized digested by thrombin and diluted in PBS. Phages are added at approximate titer of $4.23 \times 10^{12}$. Following incubation, washing, anti-M13 antibody-HRP (Cytiva Cat# 27942101, RRID: AB_2616587) is added and finally revelation is realized with ABTS substrate. The 96-well plate columns were alternatively coated by peptide/protein P53 target (P) or only blocking buffer (C for Control). Each phage is essayed simultaneously on control and protein wells. A percentage of relative signals will be calculated as: [100-C/P*100]; this gives a qualitative idea on the strength of binding between phage particles and the protein target.

## DNA sequencing and bioinformatics analysis

ELISA-positive phages are subjected to DNA extraction and sequencing using primer–96 gIII supplied in the kit. Nucleotide sequence analysis and amino acid sequence deduction were performed using the Bio-Edit Package (v.7.0.5) program.

## Molecular docking

The three-dimensional structures of p53 protein were retrieved from the RCSB Protein Data Bank [http://www.rcsb.org/pdb]. The first structure of p53 used in this study was (PDB: 2LY4. Chain B) from the unstructured transactivation domain (Residues 14–60) of the N-terminus of p53 [35]. Another structure (pdb: 3Q01. Chain A) (p53: p53 DNA-binding (Residues 94–291) and oligomerization (Residues 322–356) domain) [36] was prepared by removing water molecules while the zinc cofactors were retained in the p53 model. The structure of the peptides was constructed using the ACD/3D Viewer software.

Peptide and protein interaction patterns were predicted by Autodock Vina [37]. The grid for ligand conformational search calculations was placed with its centre located at the protein level. For the structure (2LY4 chain B), the docking grid size was 60, 40 and 50 grid points, while the grid centers were -9, 0 and 0. And for the structure (3Q01), the size of the docking grid was 60, 60 and 66 grid points, while the grid centers were designated at dimensions 35, 19 and 74. The best conformations with the lowest binding free energy were selected and analyzed using Discovery Studio 2017 R2 Client.

We should notice that a chimeric P53 protein was designed and optimized by [36] with amino acids substitutions in the DBD to reach a stabilized form. Substitutions present in the 3Q01 structure that we used are: C135V, C141V, W146Y, C182S, V203A, R209P, C229Y, H233Y, Y234F, N235K, Y236F, T253V and N268D. They are presented as such in the docking profiles.

## Supporting information

**S1 Table. Energy of docking.** a) all peptides except those of R set. b) R set representatives for 2LY4B docking. And c) R set representatives for 3Q01 docking. (Pep: peptides).
(PDF)

**S1 Fig. Docking structures of 7-mer set with 2LY4.B.** Peptides are: 7.1–7.4 (non zinc) and 7Z1-7Z4 (with zinc).
(PDF)

**S2 Fig. Docking structures of 12-mer "*Non Zinc*" set with 2LY4.B.** Peptides are 12.1–12.6.
(PDF)

**S3 Fig. Docking structures of 12-mer "*With Zinc*" set with 2LY4.B.** Peptides are 12Z1-12Z6.
(PDF)

**S4 Fig. Docking structures of PD74 set with 2LY4.B. Peptides are PD1-PD7.**
(PDF)

**S5 Fig. Docking structures of SR50 and Redundant (R) sets with 2LY4.B.** a. SR50: SR12.1 and SR12.2. b. Representatives of Redundant (R) peptides: R0-R3, R6 and R7.
(PDF)

**S6 Fig. Docking structures of *7-mer* set with 3Q01 (ribbon).** Peptides are: 7.1–7.4 (non zinc) and 7Z1-7Z4 (with zinc).
(PDF)

**S7 Fig. Docking structures of *7-mer* set with 3Q01 (interactions).** Peptides are: 7.1–7.4 (non zinc) and 7Z1-7Z4 (with zinc).
(PDF)

**S8 Fig. Docking structures of 12-mer *non zinc* set with 3Q01 (ribbon).** Representative peptides are 12.1, 12.2 and 12.6.
(PDF)

**S9 Fig. Docking structures of 12-mer *non zinc* set with 3Q01 (interactions).** Peptides are 12.1–12.6.
(PDF)

**S10 Fig. Docking structures of 12-mer *with zinc* set with 3Q01 (ribbon).** Representative peptides are 12Z1 and 1Z3-12Z6.
(PDF)

**S11 Fig. Docking structures of 12-mer *with zinc* set with 3Q01 (interactions).** Representative peptides are 12Z1-12Z3 and 12Z6.
(PDF)

**S12 Fig. Docking structures of PD74 set (control) with 3Q01 (ribbon).** Peptides are PD1-PD7.
(PDF)

**S13 Fig. Docking structures of SR50 set with 3Q01.** Peptides SR12.1 and SR12.2. Once ribbon (top) and interactions (down).
(PDF)

**S14 Fig. Docking structures of Redundant set (R) Motif 1 with 3Q01 (ribbon).** Representative peptides R0, R1 and R8-R10.
(PDF)

**S15 Fig. Docking structures of Redundant set (R) Motif 2 with 3Q01 (ribbon).** Peptides are R2-R5 and R11-R12.
(PDF)

**S16 Fig. Docking structures of Redundant set (R) Motif 2 with 3Q01 (ribbon)(continuation).** Peptides are R13-R17.
(PDF)

**S17 Fig. Docking structures of Redundant set (R) Motif 1 with 3Q01 (interactions).** Representative peptides are R0, R1 and R8-R10.
(PDF)

**S18 Fig. Docking structures of Redundant set (R) Motif 2 with 3Q01 (interactions).** Peptides are R3, R4 and R11-R13.
(PDF)

**S19 Fig. Docking structures of Redundant set (R) Motif 2 with 3Q01 (interactions)(continuation).** Peptides are R14-R17.
(PDF)

**S20 Fig. Docking profiles of « non zinc » and « with zinc » sets on DBD region of 3Q01 structures.** a. including redundant R residues. b. subtracting R residues.
(PDF)

**S21 Fig. ELISA of phages on protein P53. a)** Photo of 96-well plate after revelation and its lay-out aside. **b)** Histogram representing the responses of different tested phage clones. Wells are coated with protein P53. Phages are added separately. Following incubation with phages, anti-M13 antibody-HRP (Cytiva Cat# 27942101, RRID:AB_2616587) is added and response is revealed with HRP substrate (ABTS). Phage clones from phage display experiment against p53-derived peptides are respectively PD1 and PD7 for PD74 (12–61) and SR12.2 for SR50 (241–291).The 96-well plate columns were alternatively coated by protein P53 target (P) or only blocking buffer (C for Control). Each phage is essayed simultaneously on control and protein wells. A percentage of relative signals calculated as: [100-C/P*100]. Clones are ranked based on this percentage value representing the binding force. At secondary y axis: energies of docking on both 2LY4.B and 3Q01 structures. This ELISA is Representative.
(PDF)

**S22 Fig. Docking profiles of individual peptides.** Docking points of Redundant peptides are subtracted from the other sets to show only the "specific interactions". At left, are presented docking profiles to 2LY4.B and at right are presented docking profiles to 3Q01. (a,b) : 7 Non Zinc, (c,d) : 7 With Zinc, (e,f) : 12 Non Zinc, (g,h) : 12 With Zinc, (i,j) : PD74 recognizing set, (k,l) : SR50 recognizing set and (m,n) : Redundant (R) set.
(PDF)

**S1 File. Analysis of docking profiles of individual peptides.** a) Individual peptides docking on 2LY4.Chain B. b) Analysis of retained docking interactions in 3Q01.
(PDF)

## Acknowledgments

We thank actual and ancient members of the Laboratoire de Biotechnologie Moléculaire des Eucaryotes (LBME) mainly Azza Hadj-Sassi, Houda Skouri, Hela Trigui, Nihel Ammous, Awatef Taktak and Pr Hafedh Belguith for their valuable helps. As well as we thank members of the Unité Prestataire des Services Analytiques (UPSA) at CBS mainly Najla Fourati Masmoudi, Kamel Walha and Nessrine Kchaou for their technical assistance. Pr Sami Aifa from Laboratoire de Procédés de Criblage Moléculaire et Cellulaire (LPCMC) at the CBS is thanked for constant support and advices. Arnaud Bondon is warmly thanked for having received S. Ben Abid in his laboratory at the University of Rennes I and for his encouragement.

This publication is dedicated to the memories of Basma Hentati and Abdellatif Maalej teachers/researchers at Faculty of Sciences of Sfax.

## Author Contributions

**Conceptualization:** Sihem Ben Abid, Ines Yacoubi, Ali Gargouri.

**Data curation:** Sihem Ben Abid, Ines Yacoubi, Lamia Djemal, Salma Abdelmoula-Souissi, Aida Koubaa, Raja Mokdad-Gargouri, Ali Gargouri.

**Formal analysis:** Sihem Ben Abid, Emna Ketata, Aida Koubaa, Ali Gargouri.

**Funding acquisition:** Ali Gargouri.

**Investigation:** Sihem Ben Abid, Ines Yacoubi, Lamia Djemal, Salma Abdelmoula-Souissi, Aida Koubaa.

**Methodology:** Sihem Ben Abid, Emna Ketata, Ines Yacoubi, Lamia Djemal, Ali Gargouri.

**Project administration:** Ali Gargouri.

**Resources:** Ines Yacoubi, Lamia Djemal, Salma Abdelmoula-Souissi, Aida Koubaa, Raja Mokdad-Gargouri, Ali Gargouri.

**Software:** Emna Ketata.

**Supervision:** Ali Gargouri.

**Validation:** Sihem Ben Abid, Emna Ketata, Ines Yacoubi, Lamia Djemal, Salma Abdelmoula-Souissi, Ali Gargouri.

**Visualization:** Sihem Ben Abid, Raja Mokdad-Gargouri, Ali Gargouri.

**Writing – original draft:** Sihem Ben Abid, Emna Ketata.

**Writing – review & editing:** Sihem Ben Abid, Ali Gargouri.

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
