## [Decision Letter · Decision Letter 0]

14 Nov 2023

PONE-D-23-30568Phage libraries screening on P53: yield improvement by zinc and a new parasites-integrating analysisPLOS ONE

Dear Dr. Ben Abid,

Thank you for submitting your manuscript to PLOS ONE. After careful consideration, we feel that it has merit but does not fully meet PLOS ONE’s publication criteria as it currently stands. Therefore, we invite you to submit a revised version of the manuscript that addresses the points raised during the review process.

We look forward to receiving your revised manuscript.

Kind regards,

Asif Ali

Academic Editor

PLOS ONE

Journal Requirements:

“SBA

2 Grants

Ministry of Higher education and scientific research- Tunisia

http://www.mes.tn/index.php

NO

Center of Biotechnology of Sfax- Tunisia

http://www.cbs.rnrt.tn/

NO.”

3. We noted in your submission details that a portion of your manuscript may have been presented or published elsewhere. [We used one peptide sequence from our precedent article DOI: 10.1007/s10930-017-9730-1, which was called NG7 and here recalled 7.3. The reason is to permit to make comparison and better analysis. Ben Abid S, Sahnoun M, Yacoubi-Hadj Amor I, Abdelmoula-Souissi S, Hassairi H, Mokdad-Gargouri R, Gargouri A. New Phage Display-Isolated Heptapeptide Recognizing the Regulatory Carboxy-Terminal Domain of Human Tumour Protein p53. Protein J. 2017 Oct;36(5):443-452. doi: 10.1007/s10930-017-9730-1. PMID: 28710679.] Please clarify whether this publication was peer-reviewed and formally published. If this work was previously peer-reviewed and published, in the cover letter please provide the reason that this work does not constitute dual publication and should be included in the current manuscript.

Additional Editor Comments:

Thank you for submitting your article to PLoS ONE. We apologize for the delay. Your article has undergone review by two experts in the field, both of whom have chosen to remain anonymous. Consequently, we invite you to submit a revised version incorporating the minor revisions suggested by the reviewers.

Reviewers' comments:

Reviewer's Responses to Questions

**Comments to the Author**

1. Is the manuscript technically sound, and do the data support the conclusions?

Reviewer #1: Yes

Reviewer #2: Yes

2. Has the statistical analysis been performed appropriately and rigorously? 

Reviewer #1: Yes

Reviewer #2: Yes

3. Have the authors made all data underlying the findings in their manuscript fully available?

Reviewer #1: Yes

Reviewer #2: Yes

4. Is the manuscript presented in an intelligible fashion and written in standard English?

Reviewer #1: Yes

Reviewer #2: Yes

5. Review Comments to the Author

Reviewer #1: In the manuscript, the author has classified the target peptides for the p53 protein that plays a very important role in the cell cycle. For screening the author has used phase display and docking method where the effect of ZnCl2 observed with phase display screening and also has seen in the zinc-related peptide docking profiles. The authors describe new approach for peptide screening that may be useful for screening other targets. Here are some minor corrections in the manuscript.

1. Please correct ZnCl to ZnCl2 in whole manuscript

2. Line 88….. on whether or not ZnCl ions were………..to…… on whether or not ZnCl2/ Zn ion were added.

3. Line 90….seven times more for the 7 and 7 C-C respectively, ……to…..seven times more for the 7 mer and 7 mer C-C respectively,

4. Line 116…….We mention that sequence (NPNSAQG) that we called 7.3 appears in our previous publication…….. to ……. We mentions sequence 7.3 (NPNSAQG) which is also appears in our previous publication.

5. Line 258 …… was not suffisant enough to enhance…..to ……. was not sufficient enough to enhance

Reviewer #2: The article explores the use of phage display to isolate peptides targeting the P53 protein, a key player in cell cycle regulation and various cellular processes. The study investigates the impact of zinc supplementation during screening on the yield of plaque-forming units. Additionally, subtractive docking analyses are employed to differentiate between specific and non-specific peptide bindings, with a focus on zinc-related peptides. The results suggest advantageous docking profiles for zinc-related peptides, indicating a potential modulating role of zinc in P53 targeting.

1) The titration results in Table 1 are presented adequately, but the interpretation could be more explicit. Discuss the significance of the observed increases in phage titers, especially the substantial improvements in the zinc-treated conditions. What implications do these results have for the efficiency of the screening process?

2) While the amplification rates are presented, it would be beneficial to discuss the biological significance of these rates. What do these rates imply about the success of the screening process and the potential utility of the obtained phages?

3) The discussion of the positive effect of zinc treatment is clear but consider elaborating on why zinc was chosen for supplementation. Are there existing studies or hypotheses that support the use of zinc in phage display screenings targeting P53?

4) Discuss the differences in phage titers between eluates and amplifiates. What do these differences suggest about the role of zinc in different stages of the screening process? Are there any potential limitations or biases introduced during amplification?

5) The identification of redundant motifs is interesting but delve into the implications of these findings. What challenges do these "parasite phages" pose in terms of specificity in P53 targeting? How might these affect the reliability of the screening results?

6) Provide more context on the significance of the binding profiles in the docking analysis. How do the observed binding profiles contribute to our understanding of the interaction between the identified peptides and the P53 protein?

7) While the text touches on the effect of ZnCl on docking profiles, a more comprehensive discussion is needed. What can be inferred about the role of zinc in modulating the binding characteristics of the peptides to P53? Are there specific residues or regions that show enhanced or diminished interactions in the presence of zinc?

6. PLOS authors have the option to publish the peer review history of their article (what does this mean?). If published, this will include your full peer review and any attached files.

Reviewer #1: **Yes: **Dr. Richa Agrawal

Reviewer #2: No

---

## [Author Response · Author response to Decision Letter 0]

14 Dec 2023

Dr Sihem Ben Abid 

Laboratory of Molecular Biotechnology of Eukaryotes

Centre of Biotechnology of Sfax B.P ‘1177’’ ; 3018, University of Sfax, Tunisia

To

Editorial Manager, PLOS ONE 

Dear Asif Ali Academic Editor,

We would like through this rebuttal letter to thank you for your positive decision on our manuscript entitled “Phage libraries screening on P53: yield improvement by zinc and a new parasites-integrating analysis ” by Sihem Ben Abid, Emna Ketata, Ines Yacoubi-Hadj Amor, Raja Mokdad-Gargouri and Ali Gargouri. As well, we extend our thanks to both reviewers : Reviewer #1: Dr. Richa Agrawal and Reviewer #2. We too apologize for the delay.

Here are our responses :

 1. We ensured that our manuscript meets PLOS ONE's style requirements, including those for file naming. 

 2. Concerning the financial disclosure, we amended it as needed, by stating the following "The funders had no role in study design, data collection and analysis, decision to publish, or preparation of the manuscript." in the cover letter.

 3. Concerning the portion of our manuscript which has been published elsewhere, we provided in the cover letter some reasons that this work does not constitute dual publication and should be included in the current manuscript.

 4. We state that all data from this study are available. There are no restrictions for any reasons. The sentence « data from this study are available upon request » was just copied from a model. Here are links for the repositiory https://figshare.com/s/e94677911829f955999e. 10.6084/m9.figshare.24045519. 

 5. We included captions for our Supporting Information files at the end of our manuscript, and updated in-text citations to match accordingly.

 6.  We choose not to publish the peer review history of this article. 

Responses to Reviewers' comments:

Reviewer #1: 

1. Please correct ZnCl to ZnCl2 in whole manuscript.

 Done.

2. Line 88….. on whether or not ZnCl ions were………..to…… on whether or not ZnCl2/ Zn ion were added.

 I changed to : ZnCl2 salt was added.

3. Line 90….seven times more for the 7 and 7 C-C respectively, ……to…..seven times more for the 7 mer and 7 mer C-C respectively,

 Done.

4. Line 116…….We mention that sequence (NPNSAQG) that we called 7.3 appears in our previous publication…….. to ……. We mentions sequence 7.3 (NPNSAQG) which is also appears in our previous publication.

Changed to: We mention that sequence 7.3 (NPNSAQG) also appears in our previous publication

5. Line 258 …… was not suffisant enough to enhance…..to ……. was not sufficient enough to enhance

 Done.

Reviewer #2:

1) The titration results in Table 1 are presented adequately, but the interpretation could be more explicit. Discuss the significance of the observed increases in phage titers, especially the substantial improvements in the zinc-treated conditions. What implications do these results have for the efficiency of the screening process?

There are globally two hypothesis : either the Zn enhanced the obtention of more phages alongside the panning steps or it helped enhancing the infection of bacteria during the titration. (inserted at line 298 of the marked changes article)

Those researchers gave a rich discussion about possible explanations for this effect of metal cations (Fe, Mg and Ca) on phage infection and killing of bacteria. (inserted at line 301)

This implies that the efficiency of the screening process, in terms of representativeness, can be improved by such (ZnCl2) treatments. And, there are phage clones that may be lost at the standard method, without (ZnCl2) salt addition. (inserted at line 306)

2) While the amplification rates are presented, it would be beneficial to discuss the biological significance of these rates. What do these rates imply about the success of the screening process and the potential utility of the obtained phages?

The amplification rates reflect the multiplication of eluted phage pools by bacterial infection ; here they are almost comparable to what we can find elswhere in other screening experiments. 

We think that, in absolute, the amplification rate itself and its variation tendency through screening rounds, can not be really taken as an evaluation of the enrichment of the obtained clones or even of the success of the screening, since the redundancy (parasites) can still be recovered. And parasite clones tend to be more amplified than other clones according to literature. 

Here, the amplification rates permitted to confirm the improvement caused by zinc salt, already observed at elution. Only sequencing, sequence analysis including comparison to known parasites, and functional tests can orient us toward a potential utility of any obtained phages. (inserted at line 320)

3) The discussion of the positive effect of zinc treatment is clear but consider elaborating on why zinc was chosen for supplementation. Are there existing studies or hypotheses that support the use of zinc in phage display screenings targeting P53?

As a zinc-binding protein, P53 binds a zinc molecule and this leads to a conformation change which enables it to bind DNA. That was the main motivation to test the addition of ZnCl2 salt in screening solutions. (inserted at line 294)

Are there existing studies or hypotheses that support the use of zinc in phage display screenings targeting P53?

Already existing in the text:

However, to our knowledge no existing results about peptides targeting P53 with relation or depending on the zinc presence or state.

At our knowledge, this is the first time, zinc addition was tested in a phage display screening against P53. Results are interesting. (inserted at line 347)

By making extra literature research, we have found publications from the group (Greisman&Pabo) in which zinc finger peptides were selected by phage display, and a search for a codage between peptides residues and DNA target was attempted ; which could (theorically) inspire zinc adding. Among their articles, (Rebar et al, 1996) were suggesting adding excess of zinc in order to protect Cysteines from oxydation and to minimize any selection bias that might be caused by oxidation. (distinct aim).

Both following works (Jamieson et al, 1994) and (Choo and Klug, 1994) were cited. In the first, ZnCl2 is added to the growth medium for the expression and purification of the mature (processed form) Zinc Finger peptides as for the phage multiplication and for the band-shift assay. In the second, zinc acetate was used either in the growth medium or in zinc buffer for phage selection of Zinc Finger peptides. Similarly, (Wu et al, 1995) used ZnC12 in zinc buffer for the same selection of Zinc Fingers. 

Rebar EJ, Greisman HA, Pabo O. [8]Phage display methods for selecting zinc finger proteins with novel DNA-binding specificities. Methods Enzymol. 1996;267:129-49. doi: 10.1016/s0076-6879(96)67010-4. PMID: 8743314. 

Jamieson AC, Kim SH, Wells JA. In vitro selection of zinc fingers with altered DNA-binding specificity. Biochemistry. 1994 May 17;33(19):5689-95. doi: 10.1021/bi00185a004. PMID: 8180194. 

Choo Y, Klug A. Toward a code for the interactions of zinc fingers with DNA: selection of randomized fingers displayed on phage. Proc Natl Acad Sci U S A. 1994 Nov 8;91(23):11163-7. doi: 10.1073/pnas.91.23.11163. Erratum in: Proc Natl Acad Sci U S A 1995 Jan 17;92(2):646. PMID: 7972027; PMCID: PMC45187. 

Wu H, Yang WP, Barbas CF 3rd. Building zinc fingers by selection: toward a therapeutic application. Proc Natl Acad Sci U S A. 1995 Jan 17;92(2):344-8. doi: 10.1073/pnas.92.2.344. PMID: 7831288; PMCID: PMC42736. 

4) Discuss the differences in phage titers between eluates and amplifiates. What do these differences suggest about the role of zinc in different stages of the screening process? Are there any potential limitations or biases introduced during amplification?

Normally, amplifiates should have higher titers than eluates at each round of phage display screening (~107 times according to technical manual). Here, in the absence of zinc, both titers are lower than normal. The presence of zinc partially redressed the titers. This starts with the eluates’ titers. The amplifiates’ titers just follow, the amplification is exponential like. At elution, the effect of zinc is more remarked for the 12-mer library then for the two 7-mer libraries. Thus, a possible limitation against long protruding peptides exposed on phage pIII protein can exist, in standard conditions of screening. This is not only eliminated by zinc, but also reversed (the higher amplifiate titer). Similar limitation reversion, due to zinc presence, could concern the constrained 7-mer peptides, but more at the amplification step level. Many other sequence biases can occur during phage amplification as described by a large literature, this includes the parasite phages appearance. (Inserted at line 310)

5) The identification of redundant motifs is interesting but delve into the implications of these findings. What challenges do these "parasite phages" pose in terms of specificity in P53 targeting? How might these affect the reliability of the screening results?

Either for P53 or other targets, the possibility of identifying redundant motifs should be taken into consideration. The good study by Sloth et al [25] have identified a list of the most common "parasite sequences" that researchers have isolated sofar. They ensure that some of these sequences show functional effects on targets and thus they are not « totally » non specific. We might isolate, in a screening, either some of known parasite sequences or completely different or mixed sequences partially composed of parasites. The challenge is then to test the degree of efficency of isolated peptides and to rank them based on « specific action ». 

This heterogeneity (parasites/non parasites) probably affect the reliability of the screening results, for P53 as for other targets. Censorship and bias of the constructed libraries are thought to be the origin of these parasites. As suggested possible solutions, the high throughput sequencing and avoiding biased libraries or equilibrating biased ones. (Inserted at 363)

6) Provide more context on the significance of the binding profiles in the docking analysis. How do the observed binding profiles contribute to our understanding of the interaction between the identified peptides and the P53 protein?

The other parameter is the docking/binding profile which is a schematization that can yield informations on the interactions between the identified peptides and the P53 protein at the sequence level. This can be made either at an individual level (peptides) or for sets of peptides (to find out some common features of interaction, patterns). (Inserted at 143)

7) While the text touches on the effect of ZnCl on docking profiles, a more comprehensive discussion is needed. What can be inferred about the role of zinc in modulating the binding characteristics of the peptides to P53? Are there specific residues or regions that show enhanced or diminished interactions in the presence of zinc?

For a comprehensive view of the zinc effect in modulating the binding characteristics of the peptides to P53, we resume as follows: with the N-terminal domain docking, there is almost a superposition of binding points between the « non zinc » and the « with zinc » sets. Residues L35 and L45 are the preference of the « with zinc » set while D48 is preferred by the « non zinc ». For the C-terminal part from the docking on 3Q01 structure, the most of the binding is ubiquitous non specific. Except these close positions L350 and A353 which may be specific and the L350 preferred by the « with zinc ».

The most of the effect thus concerns the DBD ; S20 Fig was added to show the docking of the two zinc state sets on this region. Apart of some common positions between the two zinc sets, there are subregions specific to one or the other set. We notice regions 129-134, 219-223, 248-250 and 272-275 reached only by the with zinc set. The binding made by the « with zinc » set reaches frequencies higher than the non zinc set (till 30% of peptides set) indicating probably good docking points for certain positions. This concerns K132, R248, E285 (at 30%) then E271 and R273 at 20 %. Meanwhile, we notice that some « with zinc » bindings occur close to regions of redundant ubiquitous binding such as 129-134, 156-159 and 164-167. Some high frequent binding points from the « non zinc » set are absent from the « with zinc » docking, citing D148, M160, R213 and I254/T256. To conclude, we need more sequences to analyze from both conditions in order to come with stronger conclusions.

Please notice that we highlithed revisions of reviewer 1 with yellow and those for reviewer 2 with blue in the marked version of the manuscript.

Hoping that our revisions meet your expectations, the editor as well as the reviewers. And answers were not too long.

Best regards.

 Sihem Ben Abid, Corresponding authors

 Tunis (Tunisia), September 19, 2023

Ref: Ben Abid, S. et al. New Phage Display-Isolated Heptapeptide Recognizing the Regulatory Carboxy-Terminal Domain of Human Tumour Protein p53. Protein J. 2017 Oct;36(5):443-452.

---

## [Editor Report · Decision Letter 1]

3 Jan 2024

Phage libraries screening on P53: yield improvement by zinc and a new parasites-integrating analysis

PONE-D-23-30568R1

Dear Dr. Ben Abid,

We’re pleased to inform you that your manuscript has been judged scientifically suitable for publication and will be formally accepted for publication once it meets all outstanding technical requirements.

Kind regards,

Asif Ali

Academic Editor

PLOS ONE
---

## [Editor Report · Acceptance letter]

6 Feb 2024

PONE-D-23-30568R1 

PLOS ONE

Dear Dr. Ben Abid, 

I'm pleased to inform you that your manuscript has been deemed suitable for publication in PLOS ONE. Congratulations! Your manuscript is now being handed over to our production team.

Kind regards, 

on behalf of

Dr. Asif Ali 

Academic Editor

PLOS ONE